# Usefulness of circulating microRNAs miR-146a and miR-16-5p as prognostic biomarkers in community-acquired pneumonia

José María Galván-Román[1,2]*, Ángel Lancho-Sánchez[3,4], Sergio Luquero-Bueno[5], Lorena Vega-Piris[6], Jose Curbelo[1], Marcos Manzaneque-Pradales[4], Manuel Gómez[6], Hortensia de la Fuente[2], Mara Ortega-Gómez[5], Javier Aspa[3]

1 Department of Internal Medicine, University Hospital La Princesa, Madrid, Spain, 2 Department of Immunology, Biomedical Research Institute of UH La Princesa (IIS-IP), Madrid, Spain, 3 Department of Immunology, Faculty of Medicine, Universidad Autónoma de Madrid, Madrid, Spain, 4 Department of Pneumology, University Hospital La Princesa, Madrid, Spain, 5 Biobank, Biomedical Research Institute of UH La Princesa (IIS-IP), Madrid, Spain, 6 Methodology Unit, Biomedical Research Institute of UH La Princesa (IIS-IP), Madrid, Spain

* josemaria.galvan@salud.madrid.org

**Data Availability Statement:** All data generated during this research are openly available from

## Abstract

### Introduction

Patients with community-acquired pneumonia (CAP) undergo a dysregulated host response that is related to mortality. MicroRNAs (miRNAs) participate in this response, but their expression pattern and their role as biomarkers in CAP have not been fully characterized.

### Methods

A prospective observational study was performed in a cohort of 153 consecutive patients admitted to hospital with CAP. Clinical and analytical variables were collected, and the main outcome variable was 30-day mortality. Small RNA was purified from plasma of these patients obtained on the first day of admission, and miRNA expression was analyzed by RT-PCR. Univariate and multivariate analyses were carried out through the construction of a logistic regression model. The proposed model was compared with established prognostic clinical scales using ROC curve analysis.

### Results

The mean age of the patients included was 74.7 years [SD 15.9]. Their mean PSI was 100.9 [SD 34.6] and the mean modified Charlson index was 2.9 [SD 3.0]. Both miR-146a and miR-16-5p showed statistically significant association with 30-day mortality after admission due to CAP (1.10 vs. 0.23 and 51.74 vs. 35.23, respectively), and this association remained for miR-16-5p in the multivariate analysis adjusted for age, gender and history of bronchoaspiration (OR 0.95, p = 0.021). The *area-under-the-curve* (AUC) of our adjusted multivariate model (AUC = 0.954 95%CI [0.91–0.99]), was better than those of prognostic scales such as PSI (AUC = 0.799 [0.69–0.91]) and CURB-65 (AUC = 0.722 [0.58–0.86]).

Zenodo.org (https://doi.org/10.5281/zenodo.
3930832 or https://zenodo.org/record/3930832).

**Funding:** This work has been funded by the Carlos
III Health Institute (ERDF, European Regional
Development Fund), by the Spanish Society of
Pneumology and Thoracic Surgery and by the
Ministry of Science, Innovation and Universities of
Spain. The funders had no role in study design,
data collection and analysis, decision to publish, or
preparation of the manuscript.

**Competing interests:** The authors have declared
that no competing interests exist.

**Abbreviations:** AUC, Area Under the Curve; CAP,
Community-acquired pneumonia; CRP, C-reactive
protein; FC, Fold changes; IL, Interleukin; miRNAs,
Micro RNAs; PSI, Pneumonia Severity Index; ROC,
Receiver operator characteristic; RT-PCR, Real-
time Polymerase chain reaction; SIRS, Systemic
inflammatory response syndrome; TNF, Tumor
necrosis factor.

## Conclusions

High levels of miR-146a-5p and miR-16-5p upon admission due to CAP are associated with
lower mortality at 30 days of follow-up. Both miRNAs could be used as biomarkers of good
prognosis in subjects hospitalized with CAP.

## Introduction

Community-acquired pneumonia (CAP) is a frequent and severe infection. Low tract respira-
tory infections are the fifth cause of overall mortality and the first infectious cause of mortality
worldwide [1, 2]. In addition to its impact on survival, suffering from CAP affects post-episode
quality of life and functionality [3], which represents a considerable burden on the health sys-
tem [4].

Numerous strategies have been studied to improve the prediction of CAP prognosis, and
thus help in decision-making regarding the management of these patients [5]. Various widely
validated clinical scores have been developed, such as the Pneumonia Severity Index (PSI) [6]
or CURB-65 [7], capable of evaluating the clinical situation at the time of diagnosis and pre-
dicting its evolution. In parallel, several factors of the inflammatory response associated with
CAP have been studied as potential prognostic markers in these patients [8], such as procalci-
tonin, C-reactive protein (CRP) or leukocyte count, showing prognostic utility that was not
better than common clinical scales [9].

MicroRNAs (miRNAs) are small non-coding RNA molecules that have a complementary
antiparallel sequence to messenger RNAs (mRNAs). Their binding to specific mRNAs allows
post-transcriptional regulation of gene expression, blocking protein synthesis [10]. They can
be secreted to the extracellular milieu included in small extracellular vesicles called exosomes.
Through these exosomes or bound to transport proteins, miRNAs can travel in the blood-
stream and can be incorporated into other cells, thereby regulating their gene expression [11].

These molecules are very abundant, widely present in multiple tissues and biological fluids,
and have evolutionarily conserved sequences [12]. They play a role in processes such as embry-
onic development, cell death and proliferation, hematopoiesis, neurodevelopment, and meta-
bolic regulation [13, 14]. But perhaps one of their most important functions is their role in
regulating immunological processes, including the innate and adaptive immune response, the
development and differentiation of immune cells, and the prevention of autoimmune disor-
ders [15].

Regarding CAP pathophysiology, miRNAs can influence the development and function of
immune cells by blocking the translation of key proteins such as transcription factors or inter-
mediate molecules in cell receptor signaling cascades [16]. In addition, some miRNAs have
been identified as key players in modulating the immune response to severe bacterial infection,
by controlling neutrophil activation and recruitment and the chemotactic signal that initiates
the inflammatory process [17]. MiRNA determination in peripheral blood has been used for
the diagnosis of malignancies, cardiovascular diseases or autoimmune disorders [18]. More-
over, several studies have established the utility of circulating miRNAs in the diagnosis of sep-
sis [19] and in various specific infections (e.g. HIV, viral hepatitis or tuberculosis) [20].

However, there is limited scientific literature on the usefulness of these molecules as prog-
nosis markers in CAP.

In an attempt to find more accurate prognostic predictive tools for CAP, our research group set out to analyze the use of circulating microRNAs as prognostic biomarkers for mortality in this disease.

## Materials and methods

Prospective observational study in a cohort of 153 consecutive patients admitted for CAP in 2015 at a university hospital in Spain. Patients older than 18 years diagnosed with CAP in the Emergency Room were included in the study. CAP was considered when patients presented symptoms of lower respiratory tract infection together with the appearance of a new infiltrate on a chest radiograph and the absence of an alternative diagnosis during follow-up, according to the usual definition [21]. Sociodemographic and clinical variables, presence of comorbidities (individual and grouped, such as the modified Charlson index [22]), characteristics of the infectious process (including the CAP severity indices CURB65 and PSI) and analytical and radiological parameters at admission were collected. These patients underwent a blood test on the first day of admission, and were treated according to the clinical practice guidelines in force at that time [5]. The main outcome variable was 30-day mortality.

This cohort has been previously used in other studies, in the context of a larger research project on prognostic biomarkers in CAP [23, 24]. All the data generated during this research are openly available in the public repository of Zenodo.org [https://doi.org/10.5281/zenodo.3930832]. Furthermore, the methodology followed can be found in the previously published protocol [25].

## Laboratory procedures

Small RNA was purified from patients´ 250 µl plasma samples by column-based protocol, and retrotranscribed to cDNA (Exiqon's miRCURY™ series kits, 4 µl of RNA in-put); synthetic RNA controls were added in this process (spike-ins UniSP2, UniSp4 and UniSp5 before RNA extraction, and UniSP6 before retrotranscription to cDNA). After cDNA was diluted 1:40, the quality of the process was evaluated (QC control Panel) and only 117 samples passed the test (A detailed explanation of the technical criteria used for the exclusion of samples in the quality control process can be found in S1 Fig in S1 Appendix). Eight samples paired by age and gender were selected (4 patients who had suffered a cardiovascular event or death during follow-up and 4 who had not) and a panel of 752 human miRNAs was tested (miRCURY LNA™ Universal—Ready-to-Use Human Panel, Exiqon), in order to determine a preliminary pattern of differential miRNA expression between patients with different CAP progression.

According to the preliminary data obtained, 25 candidate miRNAs were selected: 4 intended to be used as normalizers, 5 selected by statistical criteria (univariate association with mortality) and 16 selected from an exhaustive bibliographic search on miRNAs, sepsis, inflammation and / or cardiovascular disease, prioritizing those that appeared in a greater number of publications and those related to respiratory diseases. RT-PCR was carried out in triplicate by hybridization with double-stranded flurochrome (ExiLENT SYBR® Green Master Mix) using the C1000 Touch CFX384 thermocycler (Bio-Rad). A PCR efficiency of 2 was assumed.

The relative amount of each miRNA was calculated with $\Delta Ct = Ct_{miRNA} - Ct_{UniSp2}$, and it was later normalized using the GeNorm algorithm and the geometric mean of the most stable miRNAs. The final data was calculated with the formula $2^{-\Delta Ct}$ and the values were expressed as the fold change (FC) of each miRNA with respect to UniSp2, as described by Marabita *et al.* [26].

Only miRNAs whose Cts were less than 2 standard deviation (SD) above the average of the least abundant spike-in, UniSp5, were taken into consideration for the analysis.

## Statistical analysis

For the descriptive analysis of the cohort, mean and SD were calculated for quantitative variables with equal variances, and median and interquartile range for quantitative variables with unequal variances. Normality of data was assessed with the Kolmogorov-Smirnov test and homoscedasticity with the Levene's test. Qualitative variables were expressed as proportion and total cases. The relationship of the different independent variables with the cumulative incidence of the main dependent variable was analyzed using the Student's t-test for quantitative variables with equal or unequal variances, or the χ2 test or the Fisher's exact test for qualitative variables, as appropriate. For the correlation analysis of the candidate miRNAs, the Pearson´s test (represented as a *heat-map*) was used, followed by the Spearman´s correlation test. Subsequently, a multivariate analysis was carried out by constructing a logistic regression model (for 30-day mortality), in order to study possible confounding and intermediate variables. All p values ≤ 0.05 were considered statistically significant, although another threshold (p ≤ 0.10) was used in the processes of variable selection, following the principle of parsimony. Selection of the most parsimonious model was made with the Likelihood-Ratio test (LR test). The predictive capacity of the estimated model as well as the comparison with established scales was made using Receiver operator characteristic (ROC) curves and subsequent comparison between areas under the curve (AUC). In addition, net reclassification index (NRI) and integrated discrimination index (IDI) were calculated. Statistical analysis was carried out using Stata v15 and R v3.5.2.

## Ethical principles

This study was previously approved by the Research Ethics Committee (REC) of Hospital Universitario de La Princesa and it was carried out following the ethical principles established in the Declaration of Helsinki, recommendations related to Good Clinical Practice, and the legislation in force regarding confidentiality. All the included patients were informed about the study and signed the informed consent, which was an inclusion criterion in this study.

## Results

A total of 153 patients were included in the study. Mean age was 75.7 years [SD 16.1], with a greater proportion of men (58.2%, n = 86). Most had a previous history of smoking (65.1%, n = 99), with a chronic obstructive pulmonary disease (COPD) prevalence of 31.4% (n = 48). The most frequent cardiovascular risk factor was high blood pressure (58.2%, n = 89), and the most frequent cardiovascular comorbidity was chronic heart failure (18.9%, n = 29). The modified Charlson index was 3.12 points [SD 2.9]. The severity of pneumonia was quantified using the usual scales: average PSI index was 103 points [SD 35.2] and average CURB-65 index was 2.78 points [SD 1.1].

Analytical and radiological variables, as well as all the prognostic scales measured were compared between surviving and deceased patients 30 days after admission. Results are shown in Table 1 and S1 and S2 Tables in S1 Appendix. Eighteen patients died in the first 30 days after admission (11.8%).

A blood sample was taken from all included patients upon admission. Small RNA was extracted from plasma samples and after quality control evaluation, only 117 samples were considered valid for miRNA analysis. A flowchart of the detailed technical criteria for exclusion of samples can be found in S1 Fig in S1 Appendix.

To assess whether sample exclusion was random, the main sociodemographic and clinical variables were compared between the group of 117 patients with valid samples and the group

**Table 1. Sociodemographic variables and comorbidities.**

| | TOTAL n = 153 | | 30-day mortality | | | | p* |
|---|---|---|---|---|---|---|---|
| | | | ALIVE n = 135 | | DECEASED n = 18 | | |
| | % | n | % | n | % | n | |
| Age—*mean / SD* | *75.68* | *16.13* | *73.82* | *16.18* | *89.69* | *5.15* | *<0.001* |
| Gender (male) | 58.17 | 89 | 58.52 | 79 | 55.56 | 10 | 0.811 |
| Ethnicity (caucasian) | 97.39 | 149 | 97.04 | 131 | 100 | 18 | 1.000 |
| *Life habits and vaccination* | | | | | | | |
| Alcoholism (active o former) | 8.50 | 13 | 8.89 | 12 | 5.56 | 1 | 1.000 |
| Tobacco use (active o former) | 65.10 | 99 | 65.93 | 89 | 58.82 | 10 | 0.563 |
| Pack-years—*mean / SD* | *27.24* | *29.66* | *27.17* | *29.37* | *27.86* | *33.38* | *0.935* |
| Pneumococcal vaccination | 41.33 | 62 | 40.60 | 54 | 47.06 | 8 | 0.611 |
| Flu vaccination (previous year) | 62.00 | 93 | 62.41 | 83 | 58.82 | 10 | 0.774 |
| *Comorbidities* | | | | | | | |
| HBP | 58.17 | 89 | 59.26 | 80 | 50.00 | 9 | 0.454 |
| DM | 16.34 | 25 | 17.04 | 23 | 11.11 | 2 | 0.739 |
| Hypercholesterolemia | 32.68 | 50 | 30.37 | 41 | 50.00 | 9 | 0.095 |
| Obesity | 42.48 | 65 | 42.96 | 58 | 38.89 | 7 | 0.743 |
| TIA | 4.58 | 7 | 3.7 | 5 | 11.11 | 2 | 0.192 |
| Stroke | 11.76 | 18 | 9.63 | 13 | 27.78 | 5 | **0.041** |
| Ischemic cardiomyopathy | 8.5 | 13 | 7.41 | 10 | 16.67 | 3 | 0.183 |
| Chronic heart failure | 18.95 | 29 | 18.52 | 25 | 22.22 | 4 | 0.750 |
| VTE | 1.96 | 3 | 1.48 | 2 | 5.56 | 1 | 0.315 |
| CKD | 14.38 | 22 | 14.81 | 20 | 11.11 | 2 | 1.000 |
| Chronic hapatopathy | 3.27 | 5 | 3.7 | 5 | 0.00 | 0 | 0.406 |
| COPD | 31.37 | 48 | 31.85 | 43 | 27.78 | 5 | 0.794 |
| Asthma | 5.23 | 8 | 5.93 | 8 | 0.00 | 0 | 0.597 |
| HIV infection | 5.88 | 9 | 6.67 | 9 | 0.00 | 0 | 0.600 |
| Solid neoplasm | 9.15 | 14 | 8.15 | 11 | 16.67 | 3 | 0.216 |
| Modified Charlson Index—*mean / SD* | *3.12* | *2.94* | *2.98* | *2.98* | *4.22* | *2.39* | *0.092* |
| *Chronic treatment* | | | | | | | |
| Bronchodilators (any type) | 33.33 | 51 | 34.07 | 46 | 27.78 | 5 | 0.791 |
| Oral corticosteroids | 3.97 | 6 | 3.73 | 5 | 5.88 | 1 | 0.518 |
| Statins | 29.14 | 44 | 28.36 | 38 | 35.29 | 6 | 0.553 |
| Antiplatelets | 26.14 | 40 | 25.19 | 34 | 33.33 | 6 | 0.460 |
| *Functional status* | | | | | | | |
| Institutionalized | 7.84 | 12 | 5.93 | 8 | 22.22 | 4 | **0.037** |
| Cognitive impairment | 17.65 | 27 | 11.85 | 16 | 61.11 | 11 | **<0.001** |
| Malnutrition | 11.11 | 17 | 9.63 | 13 | 22.22 | 4 | 0.119 |
| History of bronchoaspiration | 8.5 | 13 | 4.44 | 6 | 38.89 | 7 | **<0.001** |

CKD: Chronic kidney disease; COPD: Chronic obstructive pulmonary disease; DM: Diabetes mellitus; HBP: High blood pressure; HIV: Human immunodeficiency virus; TIA: Transient ischemic attack; VTE: Venous thromboembolism.

*Quantitative variables: Mean and SD; Qualitative variables: % and n;*

* *Qualitative variables: Chi-square test or Fisher´s exact test. Quantitative variables: t-test for equal or unequal variances as appropriate.*

of 36 patients excluded. No statistically significant differences were found (p≤0.05) (S3 Table in S1 Appendix).

Analyzing the main outcome variable among those 117 patients with a valid sample, 11 patients (9.4%) died during the 30 days after admission for CAP.

Not all the microRNAs selected as candidates for analysis were measurable with guarantees in the set of 117 patients; of the 25 candidate miRNAs, 11 were excluded from the final analysis as they were not abundant enough in one or more patients (S4 Table in S1 Appendix). Four miRNAs were used as normalizers (miR-103a-3p, miR-23b-3p, miR-23a-3p and miR -25-3p).

Association of normalized expression of each miRNA (FCs) with to 30-days mortality was analyzed (Table 2).

MiR-16-5p and miR-146a levels were both significantly higher in patients who survived compared to those who died after 30 days of follow-up (p = 0.010 and p <0.001, respectively). Distribution of these miRNAs according to mortality is shown in Fig 1.

Next, we analyzed whether the expression of both candidate miRNAs showed correlation. Heatmap representation of correlations between normalized relative quantities of the candidate miRNAs showed that miR-16-5p expression did not show a strong correlation with miR-146a, although it correlated with three other miRNAs (miR-106-5p, miR-486p and miR-144-3p; Fig 2). Moreover, a direct analysis of the correlation between miR-146a and miR-16-5p showed a weak correlation (rho = -0.57, p <0.001).

Subsequently, to assess the prognostic power of both selected miRNAs, a multivariate model was constructed through a logistic regression. For this analysis, all variables with p≤0.10 in the univariate analysis were included, as well as variables of clinical significance such as sex or comorbidity assessed by *modified Charlson* index. The PSI and CURB65 prognostic scales were excluded from the model, since they were constructed from variables already included in the multivariate analysis, and also to be able to later compare them with the fitted model.

Finally, after comparing the models using the LR test, and always keeping in the model the two significant miRNAs from the univariate analysis, the most parsimonious model was

**Table 2. miRNA relative levels according to 30-day mortality.**

| microRNAs | Total (n = 117) | | 30-day mortality | | | | p* |
|---|---|---|---|---|---|---|---|
| | | | No (n = 106) | | Yes (n = 11) | | |
| | mean | SD | mean | SD | mean | SD | |
| hsa-miR-107 | 0.17 | 0.08 | 0.17 | 0.08 | 0.19 | 0.09 | 0.343 |
| hsa-miR-17-5p | 0.67 | 0.23 | 0.69 | 0.22 | 0.56 | 0.27 | 0.074 |
| hsa-miR-21 | 3.00 | 1.42 | 2.97 | 1.45 | 3.26 | 1.05 | 0.529 |
| hsa-miR-144-3p | 4.08 | 3.98 | 4.04 | 4.06 | 4.46 | 3.17 | 0.740 |
| **hsa-miR-16-5p** | 50.19 | 40.26 | 51.74 | 41.76 | 35.23 | 14.91 | **0.010** |
| hsa-miR- 486 | 1.29 | 1.24 | 1.32 | 1.29 | 0.99 | 0.63 | 0.159 |
| hsa-miR-20a | 0.65 | 0.29 | 0.67 | 0.28 | 0.52 | 0.38 | 0.099 |
| hsa-miR-34a-3p | 0.02 | 0.03 | 0.01 | 0.03 | 0.03 | 0.05 | 0.223 |
| hsa-miR-106b-5p | 2.01 | 1.19 | 2.02 | 1.23 | 1.88 | 0.70 | 0.709 |
| **hsa-miR-146a** | 1.02 | 1.78 | 1.10 | 1.85 | 0.23 | 0.14 | **<0.001** |
| hsa-miR-483-5p | 0.02 | 0.04 | 0.01 | 0.04 | 0.04 | 0.08 | 0.285 |
| hsa-miR-125b | 0.03 | 0.06 | 0.03 | 0.04 | 0.07 | 0.17 | 0.453 |

* t-test for equal or unequal variances, as appropriate.

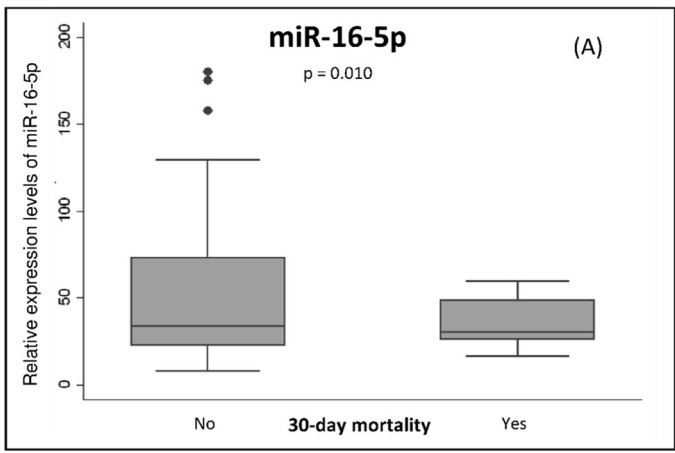

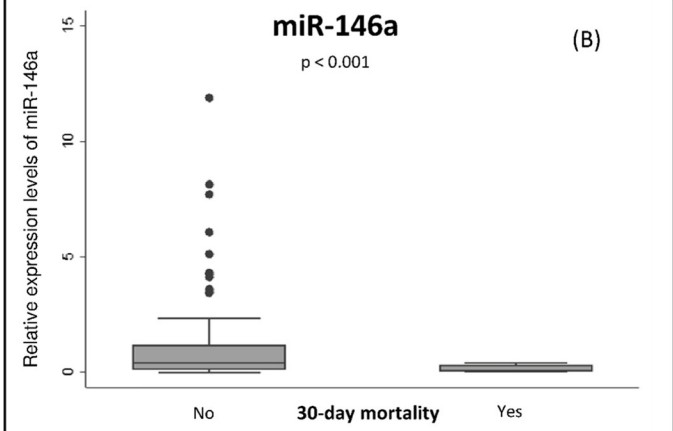

**Fig 1. Relative expression levels (FCs) at admission of miR-16-5p (A) and miR-146a (B) according to 30-day mortality after hospitalization for CAP.** Data are represented as *box plots*. Dots represent outliers. Differences were analyzed by Student´s t test for unequal variances p values ≤ 0.05 are considered statistically significant.

constructed with the variables age, sex, history of bronchoaspiration, miR-146a and miR-16-5p (Table 3).

MiR-16-5p maintained statistical significance as a prognostic marker; based on the estimated OR, it could be interpreted that for each decrease of one unit of miR-16-5p the probability of survival would increase 1.05 folds. This model presents a high LR Test ($\chi2 = 41.231$; p <0.001), and good predictive capacity ($R2_{Cox \& Snell} = 0.297$; $R2_{Nagalkerke} = 0.640$). Moreover, it is superior to the unadjusted multivariate model ($\chi2 = 14.887$; p = 0.001), with lower generalized coefficients of determination ($R2_{Cox \& Snell} = 0.119$ and $R2_{Nagalkerke} = 0.258$).

Lastly, the predictive capacity for 30-day mortality of both models and the validated prognosis clinical scales was assessed by ROC curve analysis (Fig 3).

The prognostic capacity of the adjusted model (AUC = 0.954 95%CI [0.91–0.99]), was better than the unadjusted model (AUC = 0.824 [0.73–0.92]), and likewise better than the prognostic scales PSI (AUC = 0.799 [0.69–0.91]) and CURB-65 (AUC = 0.722 [0.58–0.86]).

In addition, we sought to test how the model proposed could classify patients according to 30-day mortality compared to classic prognosis scales such as CURB-65 and PSI. For that purpose, NRI and IDI were assessed. The NRI estimated for our multivariate model vs CURB-65

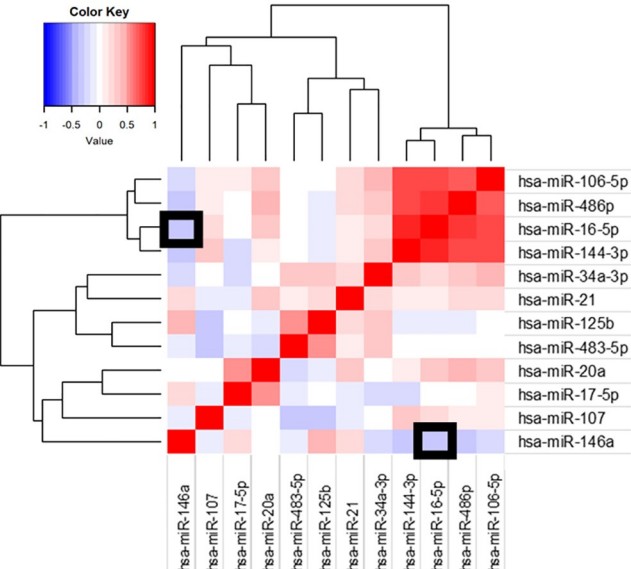

**Fig 2. Heatmap representation of correlations between miRNAs in plasma of CAP patients.** Graphic represents correlation between plasma levels of different miRNAs in CAP patients at admission. Correlation coefficients are represented by a color scale. Red colors represent positive correlations and blue colors negative correlations. Darker hues present higher coefficients. Squares outlined in black show the correlation between miR-146a and miR-16-5p.

was 59.61% (SE = 1391.69; p = 1) and vs PSI was 39.54% (SE = 1029.78; p = 1); IDI estimation for our model vs. CURB-65 was -0.40 (SE = 0.12; p = 0.007) and vs. PSI was -0.41 (SE = 0.11; p = 0.004).

## Discussion

The present study tries to assess the utility of circulating microRNA levels as prognostic biomarkers in patients admitted to hospital for CAP. For the selection of candidates, a two-step study was performed, with a first global approach using microarrays, together with a selection based on previous literature, followed by a second confirmation stage using semi-quantitative RT-PCR. After analysis of twenty-five candidate miRNAs, only two of them, miR-146a and miR-16-5p, showed a statistically significant association with mortality 30 days after admission for CAP. High levels of both miRNAs were associated with greater survival. This association

**Table 3. Multivariate models of selected miRNAs, non-adjusted or adjusted for confounding variables.**

| | NON-ADJUSTED MULTIVARIATE MODEL | | | MULTIVARIATE MODEL ADJUSTED FOR CONFOUNDING VARIABLES | | |
|---|---|---|---|---|---|---|
| | **OR** | **95% CI OR** | **p** | **OR** | **95% CI OR** | **p** |
| miR-16-5p | 0.97 | 0.94-0.99 | **0.025** | 0.95 | 0.91-0.99 | **0.021** |
| miR-146a | 0.04 | 0.00-0.81 | **0.036** | 0.05 | 0.00-2.00 | 0.109 |
| Age | | | | 1.36 | 1.05-1.77 | **0.022** |
| Sex | | | | 0.09 | 0.001-1.50 | 0.093 |
| History of bronchoaspiraton | | | | 36.49 | 1.48-899.17 | **0.028** |

Sex (female vs male), History of bronchoaspiration (yes vs no).

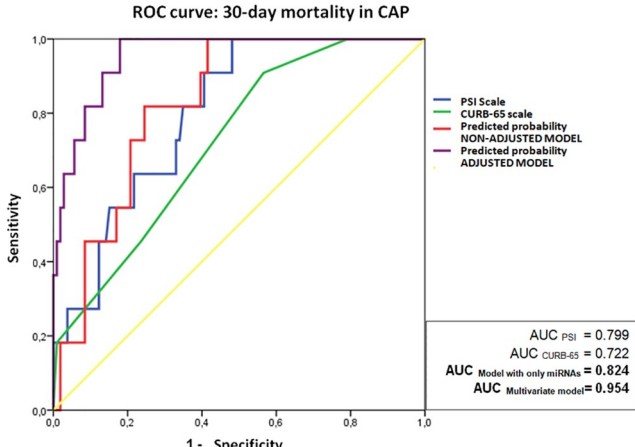

**Fig 3. ROC curve analysis of non-adjusted and adjusted multivariate models and common CAP severity scales PSI y CURB-65 for 30-day mortality.**

remained for miR-16-5p in the multivariate analysis after adjusting for age, gender, and history of bronchoaspiration. In our sample of patients admitted with CAP, this adjusted model was at least as good at predicting mortality at 30 days as the classic CURB-65 and PSI prognostic scales, after comparison of AUCs and evaluation of reclassification indices NRI and IDI. Therefore, and waiting to standardize the method and replicate it in other cohorts, our results show that the measurement of miR-146a and miR-16-5p could be useful for predicting short-term mortality after admission for CAP.

The use of circulating miRNAs as biomarkers is not new, and although is not yet widespread as routine clinical practice, it has been successfully applied in the field of respiratory diseases [27].

Regarding diagnosis, some authors have studied in depth the use of miRNAs as biomarkers for pneumonia with respect to other respiratory diseases. For this purpose miRNA levels have been determined in various biological fluids: in serum, allowing patients with pulmonary tuberculosis to be distinguished from healthy controls and patients with CAP [28]; in exosomes from pleural fluid, distinguishing between CAP and lung cancer [29]; or in sputum, discriminating active pulmonary tuberculosis from other diseases [30]. Within pneumonias, the determination of circulating miRNAs has allowed differentiating viral pneumonia from bacterial pneumonia in pediatric population [31]. It has also been used in the adult population, differentiating bacterial etiology (*Streptococcus pneumoniae*) from viral etiology (*Influenza H3N2 virus*) [32]. Interestingly, apart from studies in respiratory diseases, miR-146a determination in plasma has been successfully used as a diagnostic biomarker of sepsis in patients with clinical criteria for systemic inflammatory response syndrome (SIRS) [33].

Regarding prognosis, few studies have evaluated the ability of these molecules to predict disease progression. Wu *et al.* found that elevated miR-146a, miR-27a, miR-126, and miR-155 in serum exosomes were associated with increased occurrence of acute respiratory distress syndrome in patients with CAP; they even concluded that miR-126 could be used as a prognostic marker, as it was statistically associated with 28-day mortality [34]. In another recent article, Zhang et al., using a sepsis-specific preloaded microarray concluded that miR-223-3p could be used to predict the development of sepsis in CAP [35]. As far as we know, there are no other studies–in the literature that have investigated the use of the determination of circulating miRNAs levels to evaluate CAP prognosis.

Our finding of lower 30-day mortality in patients with elevated levels of miR-16-5p and miR-146a at admission for CAP could reflect a better inflammatory response against the invading pathogen.

MiR-16-5p has been linked to mechanisms of protection from lung damage after infection. In cell models subjected to lipopolysaccharide (LPS)-induced damage, overexpression of miR-16-5p reduced acute lung damage through inhibition of the systemic inflammatory response via inhibition of TNF-α and interleukin-6 [36]. These results have subsequently been replicated in an animal model of chronic lung infection with *Mycoplasma gallisepticum*; overexpression of miR-16-5p was able to stop the inflammatory response, exerting its inhibitory effect directly on PI3K kinase, which is a key component in the NF-κB activation cascade, and therefore, for TNF-α production [37].

Likewise, elevated miR-146a levels have been associated with reduction of LPS induced lung inflammation: exogenous addition of miR-146a significantly suppress LPS-induced inflammatory response (TNF-α, IL-6, and IL-1β expression) in alveolar macrophages, through inhibition of IRAK-1 and TRAF-6 expression, both key components of the NF-κB activation cascade [38].

Interestingly, in murine models of pneumococcal pneumonia, exogenous mimetic miRNAs that inhibit this pathway—such as miR 124 3p [39] and miR-302 [40]—promote the regeneration of alveolar epithelial cells and improve the recovery of mice affected by bacterial pneumonia.

Thus, a physiopathogenic explanation of the protective effect of circulating miR-16-5p and miR-146a observed in our patients could be related to their inhibitory effect on the inflammatory response. High levels of both miRNAs detected in CAP patients upon admission could be involved in reducing activation of the inflammatory cascades secondary to lung infection, thereby decreasing systemic inflammatory burden, and allowing a better clinical evolution in the medium term.

Furthermore, these results would be in line with those already published by our research group showing that uncontrolled inflammation in CAP and its quantification by means of blood markers allows predicting adverse prognosis in short and medium term follow-up [23, 24, 41].

We consider that the main weaknesses of our study are the difficult standardization of miRNA quantification, a common problem in this type of studies, and the exclusive recruitment of hospitalized patients, which makes it difficult to compare our miRNA data with widely used prognostic scales.

The main strengths are the sample size reached, which was sufficient to achieve statistically significant results, the use of strict quality criteria in the selection of valid samples, the use of various endogenous and exogenous miRNAs in the standardization process, and above all, the selection process of miRNAs in two steps, which ensured a good initial selection of candidates.

Nevertheless, the prognostic value of miR-16-5p and miR-146a described in this work needs to be further confirmed in routine clinical practice.

## Conclusions

In CAP patients requiring hospitalization, elevated plasma levels of miR-146a-5p and miR-16-5p measured at admission are associated with lower mortality at 30 days of follow-up. These two miRNAs could be used in the future as biomarkers of good prognosis in patients hospitalized for CAP.

## Supporting information

**S1 Appendix.**
(DOCX)

## Acknowledgments

We would like to thank Mrs. Gloria Mateos and Emilia Roy, MD, of Biomedical Research Institute La Princesa for their contribution to the revision of the Methodology section of this manuscript. We are also grateful to Fernando Moldenhauer, MD PhD, for his constant support and reviews and to Ana Gómez Berrocal, MD PhD, for her sincere opinions and the style corrections introduced in the final writing of this manuscript.

## Author Contributions

**Conceptualization:** José María Galván-Román, Sergio Luquero-Bueno, Jose Curbelo, Manuel Gómez, Hortensia de la Fuente, Javier Aspa.

**Data curation:** José María Galván-Román, Ángel Lancho-Sánchez, Sergio Luquero-Bueno, Marcos Manzaneque-Pradales.

**Formal analysis:** José María Galván-Román, Sergio Luquero-Bueno, Lorena Vega-Piris, Hortensia de la Fuente.

**Funding acquisition:** Javier Aspa.

**Investigation:** José María Galván-Román, Ángel Lancho-Sánchez, Marcos Manzaneque-Pradales, Mara Ortega-Gómez.

**Methodology:** Lorena Vega-Piris, Jose Curbelo, Manuel Gómez.

**Project administration:** Mara Ortega-Gómez, Javier Aspa.

**Resources:** Mara Ortega-Gómez.

**Software:** Lorena Vega-Piris, Hortensia de la Fuente.

**Supervision:** Sergio Luquero-Bueno.

**Validation:** José María Galván-Román, Manuel Gómez, Hortensia de la Fuente.

**Visualization:** José María Galván-Román.

**Writing – original draft:** José María Galván-Román, Lorena Vega-Piris, Javier Aspa.

**Writing – review & editing:** José María Galván-Román, Ángel Lancho-Sánchez, Manuel Gómez, Javier Aspa.

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
