## [Decision Letter · Decision Letter 0]

28 Jun 2020

PONE-D-20-09507

Usefulness of circulating miR-146a and miR-16-5p microRNAs as prognostic biomarkers in community-acquired pneumonia

PLOS ONE

Dear Dr. Galván-Román,

Thank you for submitting your manuscript for review to PLoS ONE. Your manuscript has been reviewed by three experts in the field. I agree with the comments raised by the reviewers and we feel that your study has merit, but is not suitable for publication as it currently stands. Therefore, my decision is "Major Revision”.

You must revise accordingly and explain your revisions in a covering letter if you wish for us to consider your paper further for publication. We invite you to submit a revised version of the manuscript that addresses the concerns raised by the reviewer. Please pay attention to all the reviewer suggestions and give them due consideration.

Specifically:

While the study appears well-written with overall appropriate methods, the three reviewers have raised several specific major points related to the lack of description of relevant expression and statistical values, normalization methods and use of relevant approaches to take into account the unbalanced state of the dataset. Moreover, as noted, the authors should provide the full dataset of qPCR data obtained for the primary and secondary screen as a supplemental or as a submission on a public database with all necessary information on the process.

We look forward to receiving your revised manuscript.

Kind regards,

Bernard Mari, Ph.D

Academic Editor

PLOS ONE

Journal Requirements:

Additional Editor Comments (if provided):

Reviewers' comments:

Reviewer's Responses to Questions

**Comments to the Author**

1. Is the manuscript technically sound, and do the data support the conclusions?

Reviewer #1: Yes

Reviewer #2: No

Reviewer #3: Yes

2. Has the statistical analysis been performed appropriately and rigorously? 

Reviewer #1: No

Reviewer #2: No

Reviewer #3: Yes

3. Have the authors made all data underlying the findings in their manuscript fully available?

Reviewer #1: No

Reviewer #2: No

Reviewer #3: Yes

4. Is the manuscript presented in an intelligible fashion and written in standard English?

Reviewer #1: Yes

Reviewer #2: Yes

Reviewer #3: Yes

5. Review Comments to the Author

Reviewer #1: Thank you for the opportunity to review the interesting manuscript of Galván-Román and colleagues. The study addresses an important topic of novel biomarkes for infections, and indeed miRs seem to be an appropriate target to enrich our current biomarker panels and prognostic score. One of the strengths of this study is the prospective and well-structured procedure of patient selection. Furthermore, the used methods for miR analyzation is adequate and clearly described.

In the following I listed my concerns that I would like to discuss with the authors.

Abstract:

The statement (CAP = inflammation = death) is in my optinion problematic. CAP often is associated with sepsis leading to death - thus, immunosuppression should also be considered as relevant risk factor. Therefore, I suggest rephrasing in “dysregulated” host response is more appropriate.

Please provide at least some descriptive statistics (eg. mean/median) of the relative expression when comparing survivors and non-survivors. Also providen95%-CI for the AUCs in the abstract.

Introduction:

Line 97: Improve “prognosis” or “prediction of prognosis”?

Line 124: MiRNA are also widly used as biomarkes in infections (e.g. in sepsis) I would rather cite such studies instead of referring to malignancies or AID.

Line 125: This is not true. There are several studies that already have investigeated miRs in infections! Thus, the authors should summarize the current literature of miRs and infections more appropriately.

Methods:

On the one hand the authors write about a study of 752 miR using the miRCURY LNA panel. I think the authors should include this data in the supplements. Furthermore, 25 miRs are described in the methodology, whereas only 12 are presented in the results. Please provide also these at least as supplementary materials.

Alternatively, the authors should mention that only part of the data was used for this work and the other data is evaluated separately. This should also be noted in the Data Availability statement.

In addition, no miRs that were chosen as normalizers are presented, but their prognostic miRs were normalized against UniSP2 as derived from literature. Was UniSP2 also in their data set the “best” normalizer? The authors should also provide some data describing UniSP2 as appropriate normalizer/reference (Mean with IQR + Coefficient of variation - when comparing survivors and non-survivors).

The authors should consider to correct their miR results for multiple testing.

Unfortunately, I am not able to follow the C-statics to the unadjusted and adjusted model. How are the underlying values/measures for these models composed in detail. Were new sub-scores created? How were the individual model parameters weighted?

It must be clear for the reader what this “value” is that is applied in the ROC analysis. The authors should therefore explain this in much more detail, especially because the adjusted model has a very impressive AUC, which is significantly higher than the normal AUC.

Furthermore, I suggest that the authors should consider any form of reclassification statistics to substantially show a prognostic benefit, e.g. by implementing one of the miRs (or both) in the LIS or CURB-65.

I think that these study patients were also used in different studies (e.g. citation 37-39?). If so, this need to be clarified in the methods.

Results:

Is there any association with disease severity (e.g. when comparing ICU and non-ICU patients, or correkation with LIS?).

Reviewer #2: In their study “Usefulness of circulating miR-146a and miR-16-5p microRNAs as prognostic biomarkers in community-acquired pneumonia”, the authors investigate the potential of a panel of plasma microRNAs to predict 30 day mortality of CAP patients. Upon measuring a set of miRNAs by PCR, they conclude that high levels of miR-16-5p and miR-146a-5p are associated with lower 30-day mortality.

Major Concerns:

While the authors perform thorough statistics and take the effect of confounding variables into account, they do not address the most problematic mathematical issue, which is that the dataset is hugely imbalanced. miRNAs were measured from 106 survivors vs. 11 non survivors. This has major implications for the results interpretation, as e.g. ROC curves are inappropriate to use in such a setting. A way to deal with this would be upsampling or downsampling. But given the minuscule effect size as shown in Figure 1, I doubt that the results will hold up to this. Also, the authors themselves label the dots in Figure 1 as “outliers”. I have serious reservations that without these “outliers”, the data would still be significant.

Detecting miroRNAs in plasma by PCR is an error-prone method. The authors should provide detailed information (see MIQE guidelines for reference) on the detection process.

Minor Concerns:

The study is generally well written, but some errors remain, e.g.

Line 102: procalciton should be procalcitonin

Line 281: Typo in “interestingly”

Line 285: I would refrain from using the word “evolution” in this manuscript´s context

Figure Legends need more information and axis labels need to show the type of data presented (not just "miR-146a")

Reviewer #3: This study showed high admission levels of miR-146a and miR-16-5p were significant predictors of low 30-day mortality in community acquired pneumonia (CAP). It could be a useful biomarker for prognosis of CAP. I have some major comments for improving the paper.

1. Materials and methods

- Is there any information about study period and how many patients were included, excluded and the reasons why they were excluded in this prospective study?

2. Results

-why the variables of hypercholesterolemia, stroke, cognitive impairment were not included in multivariate model as they were statistical significant in univariate analysis?

-the high standard deviation of miR-146a,miR-16-5a in alive group(table 2) indicates that the data points are spread out over a large range of values, some alive patients may have values of miR-146a,miR-16-5 within the range of the deceased group, which may not distinguish between them and what is the percentage. .

-furthermore, the value maybe also unsteady to reflect a reliable biomarkers because of possible fluctuation or emdedded extreme level in alived group . Do you have the information about the statistical analysis of the median or the trend of serial level of miR-146a,miR-16-5a between two groups, which may avoid one-point bias and became more reliable.

-Could you give the cutoff value of miRNAs in AUC model with only miRNA for predicting the 30-day mortality.

6. PLOS authors have the option to publish the peer review history of their article (what does this mean?). If published, this will include your full peer review and any attached files.

Reviewer #1: No

Reviewer #2: **Yes: **Wilhelm Bertrams

Reviewer #3: No

---

## [Author Response · Author response to Decision Letter 0]

4 Aug 2020

A new cover letter has been attached, in response to specific comments from the reviewer and editor. In addition, a separate file with the label 'Response to Reviewers' has been included.

---

## [Decision Letter · Decision Letter 1]

27 Aug 2020

PONE-D-20-09507R1

Usefulness of circulating microRNAs miR-146a and miR-16-5p as prognostic biomarkers in community-acquired pneumonia

PLOS ONE

Dear Dr. Galván-Román,

Thank you for resubmitting your manuscript to PLOS ONE. While you have adequately addressed some of the queries in the review and that the revised manuscript is significantly improved from its original submission, several critical points have not been addressed, as mainly raised by Reviewer 2.

I am sorry I cannot be more positive at the moment, but as I have noted, all is not lost. Note that it will have to go through another round of review. Please pay attention to all the reviewer suggestions and give them due consideration.

Specifically:

You should answer to the main points of Reviewer 2 regarding the method of spiking, the RNA quality data as well as a valid link pointing to the full dataset. Moreover, as proposed by Reviewer 1, the authors are encouraged to consider the opportunity of reclassification statistics to substantially show a prognostic benefit.      

We look forward to receiving your revised manuscript.

Kind regards,

Bernard Mari, Ph.D

Academic Editor

PLOS ONE

Reviewers' comments:

Reviewer's Responses to Questions

**Comments to the Author**

1. If the authors have adequately addressed your comments raised in a previous round of review and you feel that this manuscript is now acceptable for publication, you may indicate that here to bypass the “Comments to the Author” section, enter your conflict of interest statement in the “Confidential to Editor” section, and submit your "Accept" recommendation.

Reviewer #1: All comments have been addressed

Reviewer #2: (No Response)

Reviewer #3: All comments have been addressed

2. Is the manuscript technically sound, and do the data support the conclusions?

Reviewer #1: Yes

Reviewer #2: No

Reviewer #3: Yes

3. Has the statistical analysis been performed appropriately and rigorously? 

Reviewer #1: Yes

Reviewer #2: Yes

Reviewer #3: Yes

4. Have the authors made all data underlying the findings in their manuscript fully available?

Reviewer #1: Yes

Reviewer #2: No

Reviewer #3: Yes

5. Is the manuscript presented in an intelligible fashion and written in standard English?

Reviewer #1: Yes

Reviewer #2: Yes

Reviewer #3: Yes

6. Review Comments to the Author

Reviewer #1: I want to thank the authors for addressing all my raised issues. Therefore, i have no more relevant concerns.

However, I would like to encourage the authors once again to consider the opportunity of reclassification indices, e.g. NRI + IDI (as already proposed in R1). The authors themselves have pointed out that their data is prone to substantial biases needing further validation. Thus, a likely overfitted statistic along with with less robust data (as also noted by Reviewer 2) casts doubt on the validity of their results. Therefore, reclassification indices can certainly add value here.

Reviewer #2: Thank you for considering the points I raised.

I believe that, as the authors show with their statistical approach, that there are, in retrospective analysis, significant differences between the mortality groups. I do not believe that these differences have prognostic value, because the overlap between the miRNA levels of the different mortality groups is very high. On an individual patient basis, measuring these miRNAs will allow no mortality prognosis.

The model the authors derive might have its merits, though.

One thing I only noticed when reading the revised manuscript: Was the spike-in done before or after RNA purification? It reads as if it was done afterwards, and this would mean it does not reflect changes in RNA composition that are introduced during the purification procedure. If this is the case, changes in abundance might be artefacts.

Importantly, apart from citing literature and some remarks in the methods section, the authors did not provide, as I requested, data on the detection quality of the microRNAs. This plus the fact that the link to the DOI 10.5281/zenodo.3930832 returns "DOI not found" makes it still impossible to evaluate the results.

Reviewer #3: The authors have done additional great works and providing data to adequately addressed response to the comments.

7. PLOS authors have the option to publish the peer review history of their article (what does this mean?). If published, this will include your full peer review and any attached files.

Reviewer #1: No

Reviewer #2: **Yes: **Dr. Wilhelm Bertrams

Reviewer #3: No

---

## [Author Response · Author response to Decision Letter 1]

22 Sep 2020

The reply to reviewers is written in a separate document called "Response to reviewers_22_09_20".

---

## [Decision Letter · Decision Letter 2]

6 Oct 2020

Usefulness of circulating microRNAs miR-146a and miR-16-5p as prognostic biomarkers in community-acquired pneumonia

PONE-D-20-09507R2

Dear Dr. Galván-Román,

We’re pleased to inform you that your manuscript has been judged scientifically suitable for publication and will be formally accepted for publication once it meets all outstanding technical requirements.

Kind regards,

Bernard Mari, Ph.D

Academic Editor

PLOS ONE

Additional Editor Comments (optional):

Reviewers' comments:

Reviewer's Responses to Questions

**Comments to the Author**

1. If the authors have adequately addressed your comments raised in a previous round of review and you feel that this manuscript is now acceptable for publication, you may indicate that here to bypass the “Comments to the Author” section, enter your conflict of interest statement in the “Confidential to Editor” section, and submit your "Accept" recommendation.

Reviewer #1: All comments have been addressed

Reviewer #2: All comments have been addressed

2. Is the manuscript technically sound, and do the data support the conclusions?

Reviewer #1: Yes

Reviewer #2: Yes

3. Has the statistical analysis been performed appropriately and rigorously? 

Reviewer #1: Yes

Reviewer #2: Yes

4. Have the authors made all data underlying the findings in their manuscript fully available?

Reviewer #1: Yes

Reviewer #2: Yes

5. Is the manuscript presented in an intelligible fashion and written in standard English?

Reviewer #1: Yes

Reviewer #2: Yes

6. Review Comments to the Author

Reviewer #1: I have no further comments, all raised issues were addressed satisfactory within this second revision.

Reviewer #2: Thank you for addressing all my concerns.

One minor thing yet, the [] brackets around the zenodo link in the manuscript still lead to a "doi not found" error upon clicking the link, while the DOI is now in fact correct.

7. PLOS authors have the option to publish the peer review history of their article (what does this mean?). If published, this will include your full peer review and any attached files.

Reviewer #1: **Yes: **Dr T. Rahmel

Reviewer #2: No

---

## [Editor Report · Acceptance letter]

13 Oct 2020

PONE-D-20-09507R2 

Usefulness of circulating microRNAs miR-146a and miR-16-5p as prognostic biomarkers in community-acquired pneumonia 

Dear Dr. Galván-Román:

I'm pleased to inform you that your manuscript has been deemed suitable for publication in PLOS ONE. Congratulations! Your manuscript is now with our production department. 

Kind regards, 

on behalf of

Dr. Bernard Mari 

Academic Editor

PLOS ONE